# PEGylated Terpesome-Loaded 3D-Printed Aripiprazole Ocuserts for the Treatment of Ocular Candidiasis

**DOI:** 10.3390/pharmaceutics17121616

**Published:** 2025-12-16

**Authors:** Rofida Albash, Mariam Hassan, Ahmed M. Agiba, Wessam H. Abd-Elsalam, Diana Aziz, Youssef R. Hassan, Amira B. Kassem, Asmaa Saleh, Moaz A. Eltabeeb

**Affiliations:** 1Department of Pharmaceutics, College of Pharmaceutical Sciences and Drug Manufacturing, Misr University for Science and Technology (MUST), Giza 12585, Egypt; 2Department of Microbiology and Immunology, Faculty of Pharmacy, Cairo University, Cairo 11562, Egypt; mariam.hassan@pharma.cu.edu.eg; 3Department of Microbiology and Immunology, Faculty of Pharmacy, Galala University, New Galala City, Suez 43511, Egypt; 4School of Engineering and Sciences, Tecnologico de Monterrey, Monterrey 64849, Mexico; 5Department of Pharmaceutics and Industrial Pharmacy, Faculty of Pharmacy, Cairo University, Cairo 11562, Egypt; wessam.hamdy@pharma.cu.edu.eg (W.H.A.-E.); diana.aziz@pharma.cu.edu.eg (D.A.); 6Packaging Materials Department, National Research Centre (NRC), Cairo 12622, Egypt; 7Department of Clinical Pharmacy and Pharmacy Practice, Faculty of Pharmacy, Damnhour University, Damanhour 22516, Egypt; amira.kassem@pharm.dmu.edu.eg; 8Department of Pharmaceutical Sciences, College of Pharmacy, Princess Nourah Bint Abdulrahman University, P.O. Box 84428, Riyadh 11671, Saudi Arabia; asali@pnu.edu.sa; 9Department of Industrial Pharmacy, College of Pharmaceutical Sciences and Drug Manufacturing, Misr University for Science and Technology (MUST), Giza 12585, Egypt; moaz.eltabib@must.edu.eg

**Keywords:** drug repurposing, aripiprazole, PEGylated terpesomes, 3D printed ocusert, *Candida albicans* keratitis, ocular drug delivery

## Abstract

**Background/Objectives:** This study aimed to repurpose aripiprazole (AR), an antipsychotic clinically approved by the FDA, as a novel antifungal drug and to potentiate its therapeutic efficacy through PEGylated terpesomes (PEG-TERs). **Methods:** PEG-TERs were prepared by thin-film hydration and optimized using a central composite design. The optimum PEG-TER formulation was characterized for entrapment efficiency (EE%), particle size (PS), polydispersity index (PDI), and zeta potential (ZP), and subsequently integrated into polylactic acid (PLA)-based 3D-printed ocuserts. **Results:** The optimized formulation exhibited spherical vesicles with high EE%, nanoscale PS, narrow PDI, and favorable ZP, alongside excellent stability and mucoadhesive properties. *Ex vivo* permeation demonstrated a sustained release profile of AR from PEG-TERs compared with an AR suspension, while confocal microscopy confirmed efficient corneal deposition of fluorescein-labeled PEG-TERs. *In vivo*, the optimum AR-loaded PEG-TERs ocusert exhibited a substantial therapeutic effect in a rabbit model of *Candida albicans* keratitis, while histopathological assessment confirmed its ocular safety and biocompatibility. **Conclusions:** In conclusion, AR-loaded PEG-TERs embedded in PLA-based 3D-printed ocuserts represent a promising, safe, and innovative therapeutic platform for the management of *Candida albicans*-induced corneal infections, offering both drug repurposing and emerging opportunities in advanced ocular drug delivery.

## 1. Introduction

Fungal infections are an escalating threat to humans, with limited effective therapies and rapidly emerging resistance restricting treatment options [1]. These pathogens are a major cause of ocular diseases, as they can involve nearly all ocular tissues, from the eyelids and conjunctiva to the deeper intraocular structures. Among the most vision-threatening conditions are keratitis (a corneal inflammation that can lead to scarring and loss of transparency) and endophthalmitis. Among ocular fungal pathogens, *Candida* species are of particular clinical interest.

The development of novel antifungal drugs is urgently needed given the rising incidence of fungal infections, the high mortality of invasive diseases, and the limitations of current antifungal therapies [2]. Drug repurposing provides a rapid and economical therapeutic strategy to uncover antifungal potential in existing non-antifungal drugs, guided by computational modeling and validated experimentally [3]. Using this strategy, several FDA-approved drugs have shown antifungal potential, including anti-inflammatory drugs (ibuprofen, aspirin), anti-rheumatic drugs (auranofin), anticancer drugs (tamoxifen), lipid-lowering drugs (atorvastatin), and calcium channel blockers (felodipine, nifedipine) [4]. Aripiprazole (AR), an atypical antipsychotic and partial agonist of serotonin and dopamine receptors containing both piperazine and piperidine moieties, is clinically approved for the treatment of schizophrenia and bipolar disorder. Notably, it has also been reported to possess antifungal activity [5]. Although AR has recently gained attention for its promising antifungal activity, particularly against *Candida* species [5,6,7], current investigations remain limited to *in vitro* assessments and broader systemic repurposing contexts. Crucially, no topical, intraocular, or ocusert formulations of AR have been reported for managing ocular fungal infections, including ocular candidiasis. Currently, the literature lacks studies evaluating ocular delivery of AR or its therapeutic performance in models of fungal keratitis or endophthalmitis. This gap highlights a significant unmet need and reinforces the novelty of the present work, which represents the first to formulate AR within a targeted ocular nanopharmaceutical delivery system.

Nanopharmaceuticals offer a promising alternative to traditional dosage forms by improving the pharmacological efficacy of drugs while minimizing adverse side effects. AR has been successfully incorporated into various nanocarriers, including poly (caprolactone) (PCL) nanoparticles [8], niosomes [9], and nanoemulsions [10]. Terpesomes (TERs) are multifunctional vesicular carriers that provide improved penetrability compared with conventional vesicular carriers [11]. They are composed of phospholipids, surfactants, and terpenes, all of which are generally recognized by the FDA as Generally Recognized as Safe components [12]. Terpenes are generally regarded as safe for ocular use and are broadly accepted excipients. They can modulate lipid bilayer fluidity, enhancing vesicle flexibility and thereby improving drug penetration across ocular barriers [13].

Three-dimensional (3D) printing has emerged as a promising tool in pharmaceutical manufacturing [14]. In ophthalmology, it is commonly applied for both therapeutic and diagnostic purposes, including the fabrication of medical devices, as well as the development of diagnostic tools for early detection of ocular diseases [15]. In 3D printing, a digital model is first designed followed by the deposition of materials to fabricate the final product. Common 3D printing techniques include material jetting (MJ), binder jetting (BJ), selective laser sintering (SLS), and fused deposition modeling (FDM) [16]. Among these techniques, FDM is the most extensively studied, employing a nozzle-based deposition system [17]. FDM generates 3D objects by extruding melted polymeric materials through a nozzle under controlled conditions, depositing successive layers onto a designated build plate according to a computer-generated design [18]. Owing to its therapeutic customization potential, cost-effectiveness, multi-material capability, and operational simplicity, FDM has become a leading 3D printing technique in pharmaceutical research [19]. Examples of commonly used polymeric materials are acrylonitrile butadiene styrene (ABS) and the biodegradable polymer polylactic acid (PLA), which is favored for its ease of printing and biocompatibility [20].

Current evidence has revealed key aspects of the mechanisms underlying ocular surface diseases, particularly the central role of inflammation in mediating corneal damage. Corneal exposure to either microbial challenges or sterile inflammatory stimuli can trigger an immune response, leading to the recruitment of neutrophils, excessive release of proteases, and increased oxidative stress [21]. These factors collectively contribute to the degradation of the stroma and result in delayed healing. These processes underscore the significance of therapeutic strategies that not only provide antimicrobial effects but also reduce inflammatory responses at the ocular surface. In addition, several studies have shown that certain ophthalmic drugs, preservatives, antioxidants, and other formulation excipients may cause ocular irritation, epithelium disruption, or inflammatory reactions [22], underscoring the necessity for thorough assessment of ocular safety during formulation development and optimization. Understanding drug-induced inflammation is particularly relevant for repurposed drugs, where systemic safety does not necessarily predict local ocular tolerance. Furthermore, recent research has investigated the efficacy of natural bioactive compounds and biocompatible drug delivery systems to improve corneal therapy while minimizing toxicity [23,24]. These studies highlight the promising antimicrobial and protective effects of natural compounds, alongside enhanced safety profiles when integrated into complex delivery systems aimed at reducing direct epithelial contact and prolonging drug release. These findings justify the use of controlled release, biocompatible polymeric systems, such as PEGylated terpesomes (PEG-TERs) incorporated into ocuserts, to reach sufficient therapeutic levels while minimizing inflammatory adverse effects.

According to the available evidence, no previous studies have explored the use of PEG-TERs to enhance the non-invasive ocular delivery of AR. Therefore, this study aimed to develop AR-loaded PEG-TERs by incorporating Brij^®^ surfactant as an edge activator to produce highly penetrative nanovesicles, which were subsequently embedded into PLA-based 3D-printed ocuserts for controlled and sustained release. Formulation variables were optimized using Design-Expert^®^ software. Corneal penetration was evaluated through penetration studies and further visualized using confocal microscopy with fluorescein-labeled PEG-TERs. The therapeutic efficacy of the optimum PEG-TERs-embedded ocuserts was evaluated in male Albino rabbits by monitoring antifungal activity against *Candida albicans* following ocular administration. Finally, histopathological analysis was performed to confirm ocular safety.

## 2. Materials and Methods

### 2.1. Materials

Aripiprazole (AR) was obtained from Al Andalous Pharmaceutical Industries (Cairo, Egypt). L-α phosphatidylcholine, Brij^®^ 52 (HLB = 5), Brij^®^ 98 (HLB = 15), phosphotungstic acid, sodium pentane sulfonate (Na-PS), fluorescein diacetate (FDA), and mucin were purchased from Sigma-Aldrich Co. (St. Louis, MO, USA). Fenchone (a bicyclic monoterpene ketone) was purchased from Alfa Aesar (Thermo Fisher Scientific, Karlsruhe, Germany). Polylactic acid (PLA) was purchased from Flashforge Corp. (Jinhua, China) as a filament with a diameter of 1.75 mm and a print temperature of 190–220 °C. Hydrogen phosphate salts, HPLC-grade acetonitrile, acetone, and methylene chloride were all purchased from Merck KGaA (Darmstadt, Germany).

### 2.2. Methods

#### 2.2.1. Preparation of PEGylated Terpesomes

PEG-TERs were prepared by the thin-film hydration method using varying amounts of fenchone (10, 20, and 30 mg). In brief, phospholipid (100 mg), fenchone, Brij^®^ 52 or Brij^®^ 98, and AR (50 mg) were dissolved in 10 mL of methylene chloride [3]. The organic phase was evaporated under pressure at 60 °C for 30 min, using a rotary evaporator (Heidolph VV 2000, Burladingen, Germany) rotated at 90 rpm, forming a thin lipid film. The film was then hydrated with 10 mL of water [25], with continuous agitation using small glass beads for 45 min to ensure complete hydration. The dispersion was sonicated in an ultrasonic sonicator (Ultrasonicator, Model LC 60/H, Elma, Singen, Germany) for 10 min to further reduce particle size. Finally, the vesicle dispersion was stored at 4 °C.

#### 2.2.2. Characterization of PEGylated Terpesomes

##### Particle Size (PS), Polydispersity Index (PDI), and Zeta Potential (ZP)

The particle size (PS) and polydispersity index (PDI) of the PEG-TERs were measured using a Zetasizer (Malvern Instruments Ltd., Worcestershire, UK) through dynamic light scattering (DLS). The zeta potential (ZP) was determined based on the electrophoretic mobility of the vesicles [26].

##### Entrapment Efficiency (EE%)

The PEG-TERs were subjected to centrifugation (Sigma-3K30, Sigma Laboratory Centrifuges, Osterode am Harz, Germany) at 20,000 rpm for 1 h at 4 °C. The resulting supernatant was diluted and analyzed. The absorbance was measured at 255 nm [27] using a UV-Vis spectrophotometer (Shimadzu UV-1650, Kyoto, Japan).

##### Central Composite Design for Formulation Optimization

A central composite design was employed using Design-Expert^®^ software version 13 (Stat-Ease Inc., Minneapolis, MN, USA) to evaluate the effects of different variables on PEG-TERs. The design comprised 15 experimental runs. Three independent variables were evaluated: (1) lipid amount (mg, X_1_), (2) terpene amount (mg, X_2_), and (3) Brij^®^ hydrophilic-lipophilic balance (HLB) value (X_3_). The selected dependent responses were EE% (Y_1_), PS (Y_2_), and ZP (Y_3_) (Table 1).

It should be noted that only Brij^®^ 52 (HLB = 5) and Brij^®^ 98 (HLB = 15) were included in the experimental work. However, in the central composite design, the Brij^®^ HLB value (X_3_) was defined as a numeric factor from 5 to 15, and intermediate values were generated computationally by Design-Expert^®^ for model fitting and prediction. These intermediate HLB values are virtual points used for statistical interpolation rather than real surfactants.

##### Selection of the Optimum Formulation

The optimum PEG-TER formulation was selected based on the desirability function, which allows simultaneous evaluation of multiple responses. Optimization was targeted to achieve the smallest PS along with the highest EE% and ZP. The formulation exhibiting the highest desirability (closest to 1) was selected as the optimum formulation.

##### Morphological Characterization by Transmission Electron Microscopy (TEM)

The morphology of the optimum PEG-TER formulation was examined using transmission electron microscopy (TEM) (JEM-1230, JEOL, Tokyo, Japan). A drop of the sample, previously stained with 2% aqueous phosphotungstic acid, was placed on a carbon-coated copper grid [28].

##### Differential Scanning Calorimetry (DSC)

The thermal behavior of AR and the optimum lyophilized PEG-TER formulation was evaluated using an indium-calibrated differential scanning calorimeter (DSC-60, Shimadzu Corp., Kyoto, Japan).A total of 5 mg of each sample was sealed in an aluminum pan and analyzed over a range of 10 °C to 350 °C at a heating rate of 5 °C/min under a nitrogen flow of 25 mL/min.

##### Storage Stability Evaluation

The stability of the optimum PEG-TER formulation was evaluated by monitoring potential vesicle growth, drug leakage, or other physical changes during storage. The formulation was kept at 4 °C in a refrigerator for 3 months, after which PS, PDI, ZP, and EE% were measured and compared with those of freshly prepared vesicles [29].

##### Ocular Tolerance *via* pH Evaluation

The pH of the optimum PEG-TER formulation was measured at ambient temperature using a potentiometer (InoLab pH 720, WTW GmbH, Weilheim, Germany) to evaluate ocular tolerance [30].

##### *Ex Vivo* Corneal Permeation Studies

A permeation study was performed by using a Franz diffusion cell with an effective diffusion area of 0.785 cm^2^. Excised corneal tissue was carefully mounted between the donor and receptor compartments. Exactly 1 mL of AR suspension and the optimum PEG-TER formulation, each equivalent to 5 mg of AR, were placed in the donor compartment. The receptor compartment was filled with 12 mL of phosphate-buffered saline (pH 7.4) and maintained at 37 ± 0.5 °C under continuous magnetic stirring. At specific time points (1–8 h), 0.5 mL of receptor medium was withdrawn and replaced with an equal volume.

Chromatographic analysis was performed on a Hewlett Packard 1200 Series (Agilent Technologies, Waldbronn, Germany) consisting of a degasser, binary pump, autosampler, diode array detector (DAD), and ChemStation integrator. Chromatographic separations were achieved on a Luna C_18_ column (250 mm × 4.6 mm, 5.0 μm) (Phenomenex, Torrance, CA, USA) at 25 °C. Samples were injected *via* the autosampler at an injection volume of 20 µL. The mobile phase consisted of acetonitrile (B) and phosphate buffer (pH 3.0, A) containing Na-PS [31]. The permeation flux (J_max_) at 24 h and the enhancement ratio (ER) were calculated. Statistical significance was determined using an unpaired *t*-test with SPSS^®^ Statistics version 22.0 (IBM, New York, NY, USA) [32].

##### Confocal Laser Scanning Microscopy (CLSM) Evaluation

To trace the permeation of the optimum PEG-TER formulation through corneal layers, the formulation was prepared as previously described, except that the drug was replaced with fluorescein diacetate (FDA) at 1% (*w*/*v*). Freshly excised cow corneas were mounted in Franz diffusion chambers under the same conditions as the *ex vivo* corneal permeation study. FDA-loaded nanovesicles were applied to the corneal surface and left in contact for 6 h. Following exposure, the corneal tissues were embedded in paraffin wax, sectioned using a rotary microtome (Model RM2245, Leica Biosystems, Wetzlar, Germany), and evaluated for fluorescence distribution.

Visualization was carried out with an inverted confocal laser scanning microscope (Zeiss LSM710, Carl Zeiss, Oberkochen, Germany). Corneal sections were scanned under a 40× oil-immersion objective lens (EC-Plan Neofluar 40×/1.40 Oil DIC M27). Confocal images were processed using LSM Image Browser, release 4.2 (Carl Zeiss MicroImaging GmbH, Jena, Germany) [33].

##### Formulation and 3D Printing of Ocuserts

Before manufacturing the ocusert, the mechanical characteristics of the PLA filament were assessed by using a texture analyzer. The tensile strength of the PLA filament was determined according to the ASTM D638-03 standard [ASTM D638-03] using a Zwick/Roell Z2.5 testing machine (Zwick/Roell GmbH & Co. KG, Ulm, Germany) fitted with a 2.5 kN load cell and operated at a crosshead speed of 40 mm/min. Data acquisition and processing were performed with testXpert II testing software (version 3.2).

The ocusert was designed using 3D CAD software (Rhinoceros 3D, Version 5.14, McNeel & Associates, Seattle, WA, USA) and fabricated with an FDM 3D printer (CoLiDo Plus 2.0 system, Hong Kong, China) using Cura slicing software (Version: 5.10.2). The printing parameters were set as follows: extrusion temperature, 210 °C; bed temperature, 60 °C; printing speed, 40 mm/s; nozzle diameter, 0.4 mm; layer height, 0.15 mm; layer width, 0.48 mm; and infill percentage, 20%. Cooling fans were automatically operated during the printing process.

The design of the 3D-printed PLA ocusert was based on a prototype reported in a previous study by one of the authors, which described 3D-printed PLA inserts laden with ultra-fluidic glycerosomes for the management of ocular cytomegalovirus retinitis [20]. In that study, the mechanical characteristics, flexibility, comfort, and ocular compatibility of the PLA-based inserts were rigorously evaluated and verified for safe ocular administration and efficient drug delivery. By following the same structural design parameters, including thickness, geometry, flexibility, and polymer composition, the current ocusert maintains ocular compatibility.

The graphical design of the ocusert, including its dimensions, is illustrated in Figure 1a–c and the 3D printing setup is shown in Figure 1d. The lower lid (Figure 1a) was designed with an internal cavity to accommodate the drug formulation, whereas the upper lid (Figure 1b) was designed as a flat cover. The 3D CAD representation of the assembled ocusert is shown in Figure 1c. For assembly, the drug cargo (20 µL) was loaded into the lower lid, and the upper lid was secured over it using PLA-acetone (0.2% *w*/*v*) [20]. The assembled ocuserts were dried at 25 °C inside a desiccator and subsequently stored in airtight, sealed containers at 6 ± 2 °C for further studies. The final dimensions of the assembled ocuserts were measured using a micrometer caliper (Mitutoyo, Kawasaki, Japan).

##### *In Vitro* Antifungal Efficacy

*Candida albicans* (ATCC 60193) was used as the test organism in all experiments [34]. The minimum inhibitory concentration (MIC) of AR was determined using the broth microdilution method, according to the guidelines of the Clinical and Laboratory Standards [35].

##### *In Vivo* Ocular Evaluation

All animal experiments and technical procedures were approved by the Research Ethics Committee (REC) of the Faculty of Pharmacy, Cairo University (Approval No.: MI3692, 30 September 2025). Six adult male Albino rabbits with a weight of 2.5 ± 0.2 Kg were housed individually at controlled conditions (25 ± 2 °C; 12 h light/12 h dark cycle) with free access to standard commercial food and tap water *ad libitum*. The rabbits were randomly divided into 2 groups; group I (*n* = 3) received ocusert AR suspension and group II (*n* = 3) received ocusert optimum PEG-TER formulation. The sample size (*n* = 3 per group) was selected based on ethical reduction principles and is consistent with exploratory preclinical ocular studies.

The experiment was performed as previously described [35]. In brief, the ocusert-loaded AR suspension and ocusert-loaded optimum PEG-TER formulation were inserted into the lower conjunctival sac of the right eye of each rabbit using a micropipette. The left eye served as a control (no drug administered). At specific time points (1–8 h), 4 sterile filter paper discs (Whatman Inc., Florham Park, NJ, USA) were placed under the eyelid of each eye to collect tear fluid. For each eye (right and left eyes), 2 discs were placed in a 1.5 mL Eppendorf tube containing 500 μL Sabouraud dextrose broth (SDB) inoculated with a 10% (*v*/*v*) yeast suspension (10^7^ CFU/mL). The remaining 2 discs were placed in a 1.5 mL Eppendorf tube containing 500 μL uninoculated SDB, which served as blanks for optical density (OD) measurements. After incubation, 200 μL from each tube was transferred to a sterile 96-well plate, and the optical densities (OD_600nm_) were measured by using an automated spectrophotometric plate reader (Synergy 2, Biotek, Winooski, VT, USA) [34]:(1)$$Growth\;inhibition\left(\%\right)=\frac{Control\left(left\;eye\right){OD}_{600nm}-Test\left(right\;eye\right){OD}_{600nm}}{Control\left(left\;eye\right){OD}_{600nm}}\times 100$$

##### Histopathological Examination

At the end of the experiment, 3 rabbits from each group were sacrificed for histopathological examination. The ocular tissues were excised and preserved in 10% formalin and then processed. Thin sections were prepared and stained with hematoxylin and eosin stain [13].

## 3. Results and Discussion

### 3.1. Optimization of PEG-TERs via Central Composite Design

A quality by design approach was adopted to optimize the formulation of PEG-TERs. This systematic approach promotes efficient product development by deepening process understanding, boosting formulation performance, and accelerating overall product development. Model fitting demonstrated that EE% was best fitted by a quadratic model, while PS and ZP followed linear models. The adequacy of the models was verified through adequate precision, with all responses showing values above 4 (Table 2), indicating acceptable signal-to-noise ratios. Furthermore, the design analysis showed close agreement between the predicted and adjusted R^2^ values, confirming the robustness of the models.

### 3.2. Effect of Formulation Variables on EE%

The response surface plot (Figure 2a) showed that both lipid and terpene amounts significantly affected the EE% of AR-loaded PEG-TERs. Increasing the lipid amount was associated with a gradual increment in EE%, likely due to the greater availability of the lipid matrix for accommodating and stabilizing the active drug [36]. In turn, higher amounts of terpenes led to a decrease in EE%, possibly due to their solubilizing effect on the drug and their interference with lipid packing [37], which consequently promotes drug leakage. These findings emphasize the importance of balancing the lipid-to-terpene ratio to achieve maximum drug entrapment, consistent with the optimization outcomes of the central composite design. The EE% of the formulated PEG-TERs ranged between 66.27 ± 0.07% and 99.05 ± 0.05% (Table 3), indicating high EE% across most terpesomal formulations and demonstrating the effectiveness of the design approach. Such elevated EE% values can be attributed to the strong affinity of AR (as a lipophilic molecule) for the hydrophobic domains of the lipid–terpene bilayer, which promotes its incorporation and retention within the vesicular matrix.

### 3.3. Effect of Formulation Variables on PS

Figure 2b showed a direct relationship between lipid amount and PS, where increasing the lipid amount resulted in a gradual enlargement of PS [38]. This effect may result from the enhanced viscosity of the dispersion medium at higher lipid amounts, which can hinder efficient particle disruption and promote the formation of larger vesicles. On the other hand, variations in terpene amount showed a relatively minimal effect on PS, indicating that lipid amount was the predominant factor governing PS. These findings suggest that careful adjustment of lipid content is critical for maintaining PS within the nanometric range to ensure favorable ocular delivery. In the current study, the PS values of PEG-TERs ranged between 228.14 ± 3.00 nm and 565.94 ± 5.00 nm (Table 3), confirming that all terpesomal formulations remained within a size range suitable for ocular administration.

### 3.4. PDI Evaluation

The PDI reflects the width of the PS distribution within unimodal nanoparticle populations [39]. A PDI value of 0 indicates a homogeneous dispersion, while a value approaching 1 indicates a heterogeneous system [26]. The PDI values of the formulated AR-loaded PEG-TERs ranged between 0.431 ± 0.025 and 0.874 ± 0.005 (Table 3), indicating a moderate to broad PS distribution depending on the formulation composition; subsequent thin-film hydration and sonication resulted in high PDI values [40].

### 3.5. Effect of Formulation Variables on ZP

The response surface plot for ZP (Figure 2c) showed that both lipid and terpene amounts influenced the surface charge. Increasing the lipid amount was associated with a shift toward more negative ZP values, which could be due to the presence of ionizable phosphate groups within the lipid matrix that enhance surface charge density toward negative charge [41]. In turn, higher terpene amounts reduced the magnitude of ZP, likely due to their intercalation within the lipid bilayer, which disrupts charge distribution and compromises vesicle stability [42]. Since higher absolute ZP values are often associated with improved colloidal stability, these findings emphasize the importance of optimizing lipid-to-terpene ratios to balance drug permeability with formulation stability. In the current study, the ZP values of AR-loaded PEG-TERs ranged between −30.80 ± 0.50 mV and −41.28 ± 0.50 mV (Table 3), which fall within the range typically considered suitable for stable nanovesicles, as the relatively high negative surface charge provides sufficient electrostatic repulsion between vesicles, effectively preventing aggregation and promoting a stable nanosystem.

### 3.6. Selection of the Optimum AR-Loaded PEG-TERs

The optimum PEG-TER formulation was selected using Design-Expert^®^ software. The selected formulation achieved a desirability value of 0.785 (Figure 2d). This optimum terpesomal formulation (F14) (Table 3) was prepared using Brij^®^ 98 (HLB = 15), 150 mg of lipid, and 10 mg of terpene, resulting in an EE% of 99.05 ± 0.05%, a PS of 447.15 ± 5.00 nm, a PDI of 0.742 ± 0.008, and a ZP of −34.70 ± 0.30 mV. These findings confirm that the optimum formulation fulfilled the design criteria and demonstrated suitability for effective ocular delivery.

### 3.7. Morphological Evaluation by TEM

From Figure 3, the nanovesicles demonstrated a nearly spherical shape with defined boundaries and smooth surfaces. The observed PS distribution was consistent with the nanometric range (228.14 ± 3.00 nm and 565.94 ± 5.00 nm) detected by DLS analysis, confirming the successful formation of uniform terpesomal nanovesicles. Moreover, the smooth surface texture and absence of visible clumping further indicate satisfactory colloidal stability, which is consistent with the high negative zeta potential values reported for these formulations. Such morphological characteristics are advantageous for ocular delivery, as spherical, uniformly dispersed vesicles promote enhanced corneal interaction and improved spreading over the ocular surface.

### 3.8. Thermal Evaluation by Differential Scanning Calorimetry (DSC)

In the thermogram of AR (Figure 4a), a characteristic endothermic peak was observed, corresponding to its melting point and reflecting the crystallinity of the AR [43]. This sharp thermal peak indicates that AR exists in a highly ordered crystalline form at 140–148 °C. In turn, the thermogram of the optimum AR-loaded PEG-TER formulation (Figure 4b) showed the disappearance (or marked broadening) of the AR melting endotherm, suggesting that the drug was either molecularly dispersed or transformed into an amorphous state. This transformation is advantageous, as amorphous or molecularly dispersed drug forms generally exhibit enhanced solubility and improved release characteristics compared with crystalline counterparts. Furthermore, the absence of a distinct AR peak confirms successful entrapment of AR within the lipid bilayer. To conclude, the DSC results indicate that AR was efficiently incorporated into the terpesomal vesicular system, leading to a more thermodynamically stable and molecularly homogeneous nanoformulation.

### 3.9. Short-Term Storage Stability

After 3 months of storage at 4 °C, the physical form of the optimum PEG-TER formulation remained unchanged. The physicochemical properties of the stored samples were statistically compared with fresh samples, and no significant differences were detected in EE% (98.50 ± 1.20%), PS (451.21 ± 3.12 nm), PDI (0.702 ± 0.001), or ZP (–35.89 ± 1.12 mV) (*p* > 0.05). This high stability may be ascribed to the existence of phospholipid (L-α phosphatidylcholine) and PEGylated surfactant (Brij^®^ 98). PC forms a robust bilayer structure that contributes to membrane integrity and minimizes vesicle fusion, while also providing a natural barrier that resists environmental stress. The high HLB value of Brij^®^ 98 enhances the hydrophilicity and steric stabilization of the vesicle surface, thereby minimizing aggregation and improving dispersion stability [44].

### 3.10. Ocular Tolerance Testing (pH Evaluation)

Maintaining a pH within the physiological range is essential to minimize ocular irritation and ensure patient comfort during ocular administration [35]. The pH of the optimum PEG-TER formulation was found to be at 7.0 ± 0.02, which is close to the physiological tear fluid pH (7.4), thereby indicating their suitability for ocular drug delivery [35].

### 3.11. Ex Vivo Corneal Permeation Evaluation

Encapsulation of AR within PEG-TERs resulted in enhanced drug retention and reduced permeation compared with AR in suspension form (Figure 5). This was confirmed by lower Q_24_ and J_max_ (*p* < 0.05) (Table 4). Ingredients, PC and terpenes, were reported to enhance drug localization while decreasing transmembrane penetration. PC, due to its hydrophobic nature, facilitates drug deposition and retention inside vesicular membranes [45]. Moreover, fenchone has a high boiling point of 192 °C. Elnabrawi and his colleagues reported that terpenes with higher boiling points demonstrate stronger molecular cohesiveness, leading to increased viscosity and, consequently, a slower drug diffusion profile [46]. This effect is particularly advantageous for ocular delivery, where prolonged residence time and reduced dosing frequency are essential for improving therapeutic outcomes and patient adherence.

### 3.12. Corneal Permeation Evaluation by CLSM

CLSM was used to evaluate the infiltration efficiency, as well as the fluorescence distribution and intensity following ocular administration of fluorescein-labeled PEG-TER formulation. The CLSM image (Figure 6) showed that the FDA-loaded PEG-TER formulation demonstrated intense fluorescence, broadly distributed across multiple ocular layers. Longitudinal sections further confirmed the penetration depth of the nanovesicles within corneal tissues, showing a uniform diffusion pattern. These findings are consistent with the *ex vivo* ocular permeation results and may be explained by the reduced PS of the PEG-TER formulation, which facilitates their penetration into ocular tissues.

### 3.13. In Vitro Antifungal Activity

AR showed antifungal activity against *Candida albicans*, with a MIC of 5 mg/mL. The antifungal efficacy of the optimum PEG-TER formulation incorporated into a PLA-based 3D-printed ocusert was compared with that of an AR suspension-loaded ocusert. The PEG-TER-loaded ocusert demonstrated significantly greater antifungal activity than the AR suspension-loaded ocusert throughout the 8 h-study period of the post-treatment (*p* < 0.05). For the AR suspension-loaded ocusert, antifungal activity peaked at 39.1 ± 5.4% after 1 h, then declined gradually, remaining above 20% inhibition at 3 h and decreasing further to 5.5 ± 1.7% after 6 h (Figure 7). In turn, the PEG-TER-loaded ocusert showed a gradual increase in antifungal activity, reaching a maximum of 64.7 ± 6.9% at 3 h, and maintaining high inhibition (64.7–47.7%) up to 4 h post-treatment (Figure 7). The calculated AUC_0–8_ for the PEG-TER-loaded ocusert (307.7 ± 13.1) was significantly higher than that of the AR-loaded ocusert (100.5 ± 13.7), confirming the superior and sustained antifungal performance of the PEG-TER formulation. The enhanced antifungal activity of the PEG-TER-loaded ocusert can be due to several formulation-related factors. Encapsulation of AR within PEG-TERs improved its overall therapeutic efficacy by enabling a more controlled and sustained release from the 3D-printed PLA matrix. The terpesomal structure, composed of PC and Brij^®^ 98 surfactant, provided both enhanced membrane interaction and sustained drug retention at the site of action. In addition, the presence of fenchone, a terpene with moderate lipophilicity and high molecular cohesiveness, likely promoted better corneal and fungal cell wall permeation, resulting in higher intracellular accumulation of AR [47]. All these factors contributed to the significantly higher AUC_0–8_ and prolonged antifungal effect observed for the PEG-TER-loaded ocusert compared with the AR suspension-loaded counterpart.

### 3.14. Histopathological Evaluation

Histopathological evaluation of corneal tissues showed variations among the treatment groups. The untreated control group preserved a normal histological architecture of both the epithelium and stroma, with no detectable alterations (Figure 8a). Corneas from rabbits treated with AR suspension-loaded ocuserts demonstrated mild vacuolation of the corneal epithelium along with a high density of new blood vessels (Figure 8b). In turn, those treated with blank ocuserts exhibited severe vacuolation of the corneal epithelium, accompanied by a few newly formed blood vessels and mild stromal edema (Figure 8c). On the other hand, corneas from rabbits treated with PEG-TER-loaded ocuserts showed moderate vacuolation in the corneal epithelium, while the stromal structure remained largely conserved (Figure 8d). Collectively, these findings suggest that PEG-TER-loaded ocuserts exhibited better ocular tolerance compared with AR suspension- and blank-loaded ocuserts, with histological topographies more closely resembling those of the untreated control.

### 3.15. Future Perspectives and Benefits of the Research

The findings of this study open several avenues for future research and clinical translation. First, the successful repurposing of AR as an antifungal agent highlights the broader potential of screening existing drugs for ocular infectious conditions or ocular pathogens, offering a faster and more cost-effective strategy or approach compared with traditional antifungal discovery. Second, the PEG-TERs platform may be further engineered to administer or deliver combination therapies or adjuvants that target biofilm formation or antifungal resistance pathways, potentially enhancing therapeutic outcomes in recalcitrant fungal keratitis. In addition, integrating PEG-TERs into 3D-printed PLA ocuserts demonstrates the feasibility of customizable, patient-specific ocular implants, which could be tailored in terms of drug content, structural design, and release kinetics using additive manufacturing technologies. Future work may also explore the long-term stability, scalability, and regulatory considerations of such hybrid nano-3D systems to facilitate their transition into clinical practice. Overall, this technology presents a promising foundation for next-generation sustained-release ocular drug delivery systems aimed at improving therapeutic precision, patient compliance, and treatment success in fungal and potentially other infectious or inflammatory ocular conditions.

## 4. Conclusions

This study successfully repurposed AR, an FDA-approved antipsychotic, as a novel antifungal candidate and enhanced its therapeutic performance through formulation into PEG-TERs. The optimum formulation showed high EE%, nanoscale size, uniform distribution, favorable surface charge, and excellent physical stability, and was further incorporated into PLA-based 3D-printed ocuserts. The developed ocuserts offered controlled and sustained drug release, efficient corneal deposition, and potent antifungal activity against *Candida albicans*, as confirmed by *ex vivo* and *in vivo* experiments. Importantly, histopathological analysis verified the ocular safety and biocompatibility of the optimum hybrid system. Collectively, AR-loaded PEG-TERs ocuserts represent a safe, effective, and emerging platform for the management of *Candida albicans*-induced keratitis, emphasizing the potential of drug repurposing and 3D printing technologies in advanced ocular drug delivery and controlled release.

## Figures and Tables

**Figure 1 pharmaceutics-17-01616-f001:**
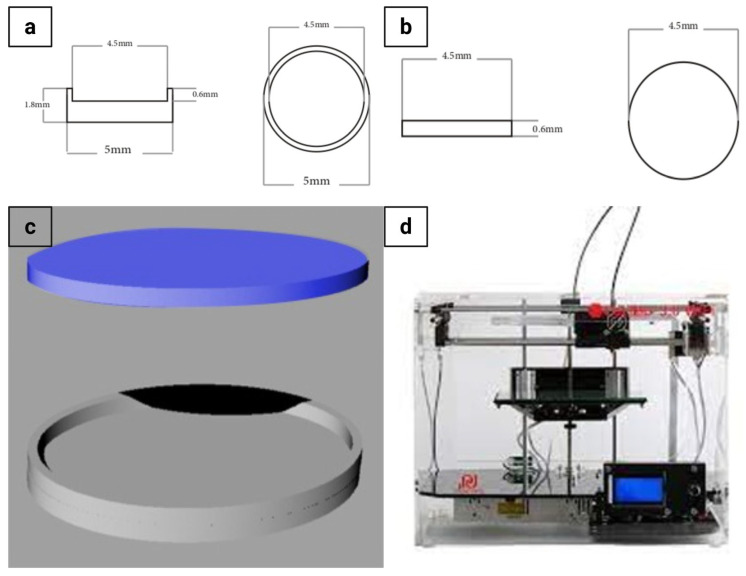
Graphical design and 3D printing of the ocusert device. (**a**) Schematic top and side views of the lower ocusert lid showing detailed dimensions. (**b**) Schematic top and side views of the upper ocusert lid. (**c**) 3D CAD model illustrating the ocusert assembly (upper and lower lids). (**d**) FDM 3D printer used for fabrication.

**Figure 2 pharmaceutics-17-01616-f002:**
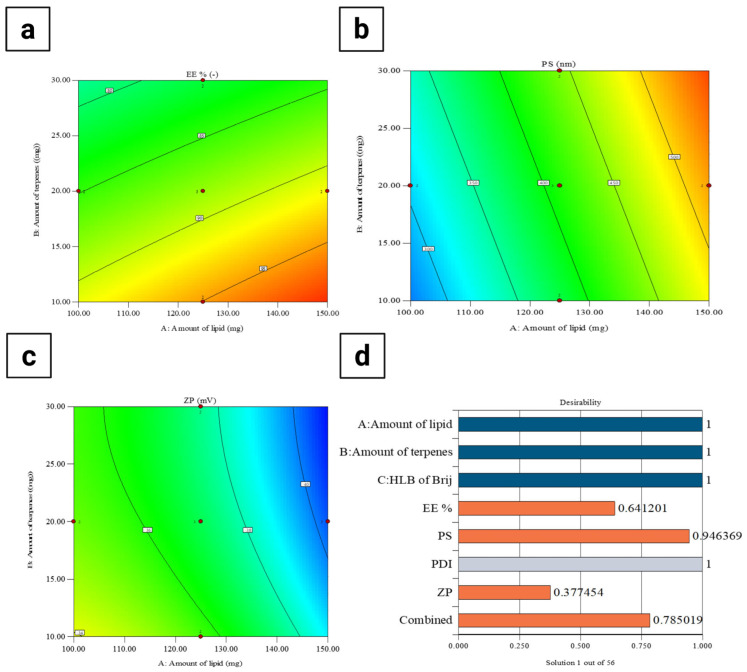
Response surface plots of formulation variables affecting (**a**) EE%, (**b**) PS, (**c**) ZP, and (**d**) desirability of AR-loaded PEG-TERs.

**Figure 3 pharmaceutics-17-01616-f003:**
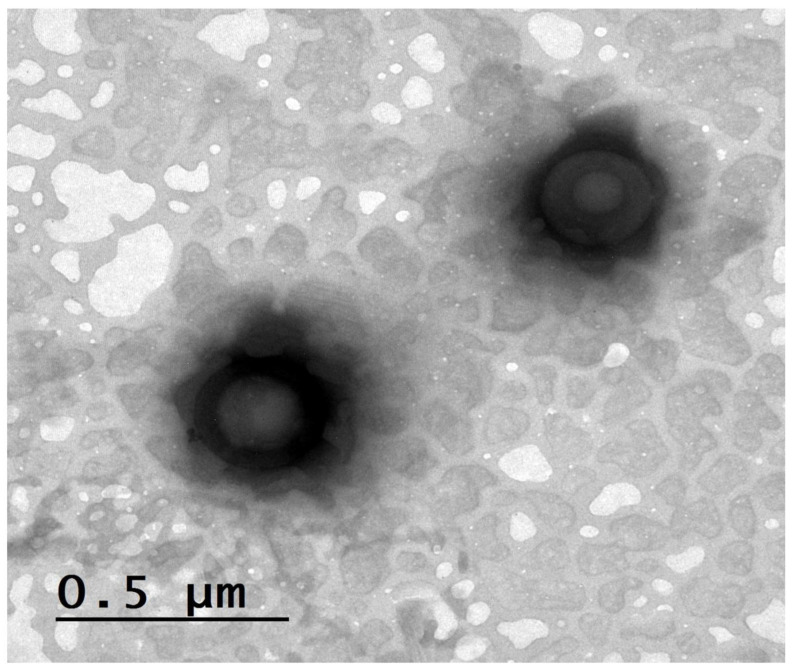
TEM image of AR-loaded PEG-TERs showing their morphological structures.

**Figure 4 pharmaceutics-17-01616-f004:**
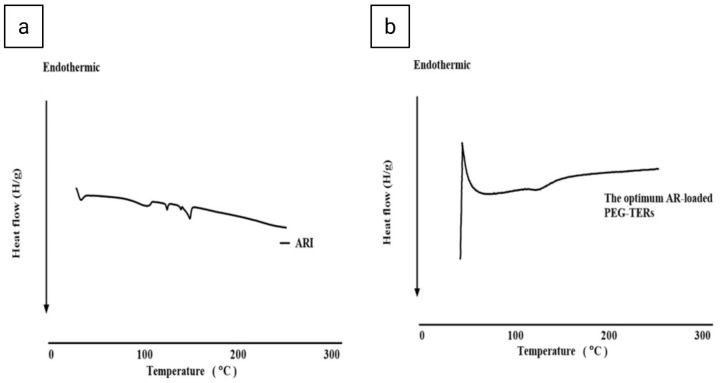
Differential scanning calorimetry thermograms of (**a**) AR and (**b**) the optimum AR-loaded PEG-TER formulation.

**Figure 5 pharmaceutics-17-01616-f005:**
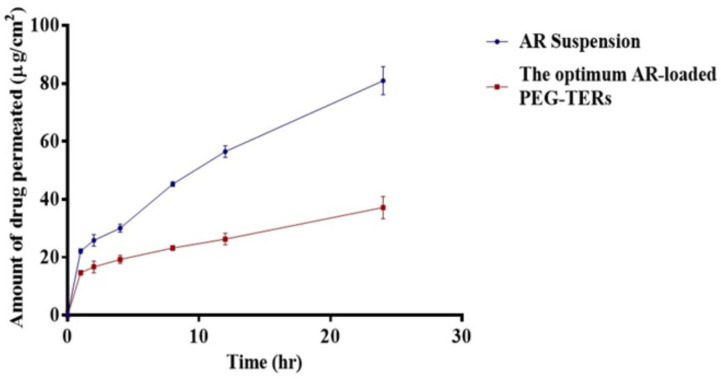
*Ex vivo* Corneal Permeation of AR suspension and the optimum AR-loaded PEG-TER formulation.

**Figure 6 pharmaceutics-17-01616-f006:**
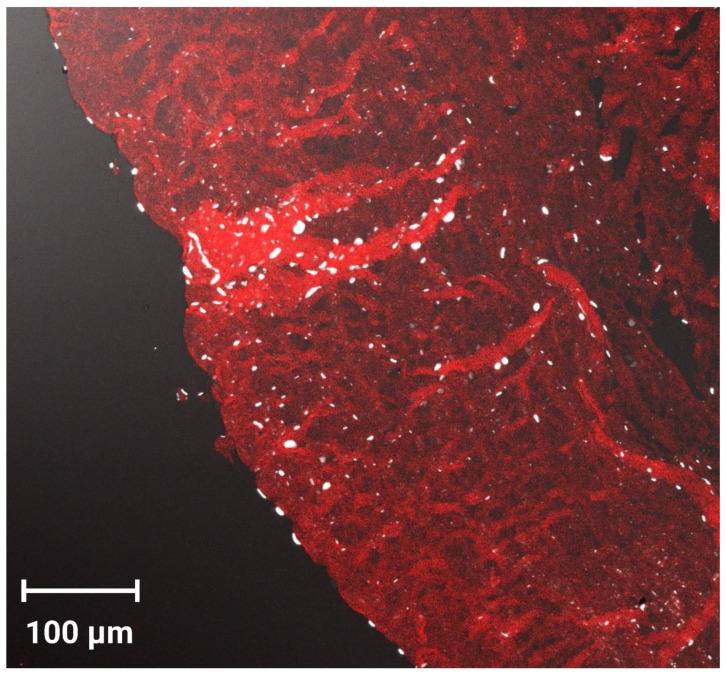
Tile scan confocal laser microscope micrographs of cornea treated with the FDA-loaded PEG-TER formulation.

**Figure 7 pharmaceutics-17-01616-f007:**
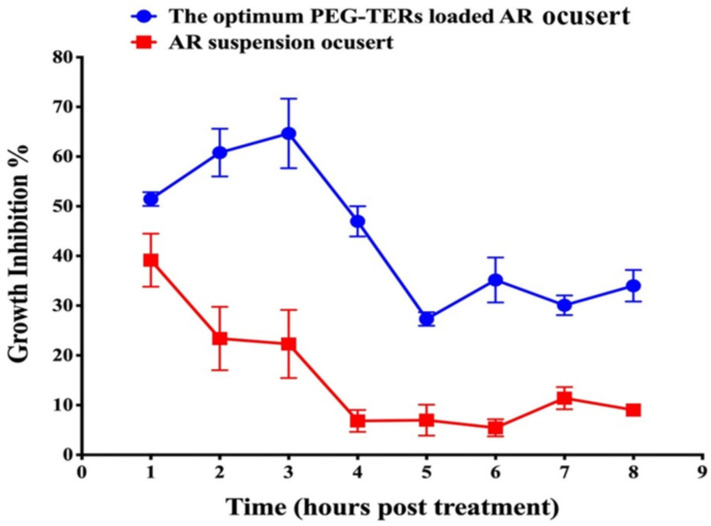
Comparative antifungal activity of PEG-TER-loaded and AR suspension-loaded 3D-printed ocuserts against *Candida albicans*.

**Figure 8 pharmaceutics-17-01616-f008:**
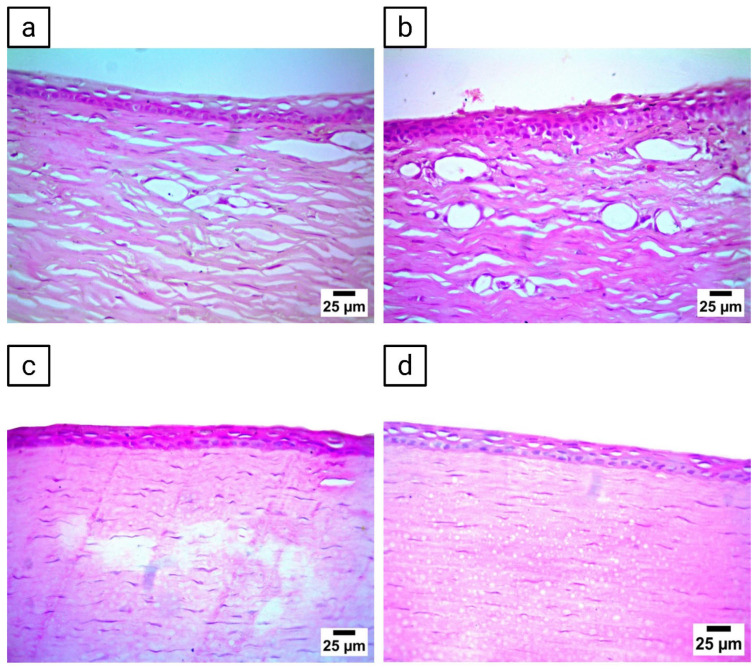
Histopathological evaluation of corneal tissues following ocular administration of different formulations. (**a**) Cornea from the untreated control group. (**b**) Cornea from rabbits treated with AR suspension-loaded ocusert, (**c**) Cornea from rabbits treated with blank ocusert, and (**d**) cornea from rabbits treated with PEG-TER-loaded ocusert.

**Table 1 pharmaceutics-17-01616-t001:** Central Composite Design for AR-Loaded PEG-TERs.

**Factors** **(Independent ariables)**	**Factor type**	**Levels**		
		**(−1)**		**(+1)**
X_1_: Amount of lipid (mg)	Numeric	100		150
X_2_: Amount of terpenes (mg)	Numeric	10		30
X_3_: Brij^®^ HLB value	Numeric	5		15
**Responses** **(Dependent variables)**		**Constraints**		
Y_1_: EE%		Maximize		
Y_2_: PS (nm)		Minimize		
Y_3_: ZP (mV)		Maximize (Absolute value)		

**Table 2 pharmaceutics-17-01616-t002:** Central Composite Design for Optimization of AR-Loaded PEG-TERs.

Responses	R^2^	Adjusted R^2^	Predicted R^2^	Adequate Precision	Significant Factors
EE%	0.89	0.87	0.83	19.80	X_1_, X_2_, X_3_
PS (nm)	0.90	0.88	0.87	29.93	X_1_, X_2_, X_3_
ZP (mV)	0.91	0.88	0.81	16.64	X_1_, X_2_, X_3_

**Table 3 pharmaceutics-17-01616-t003:** Results of the Central Composite Design for AR-Loaded PEG-TERs.

Formula Code	Lipid Amount (mg)	Terpene Amount (mg)	HLB of Brij^®^	EE (%)	PS (nm)	PDI	ZP (mV)
**F1**	100	10	5	94.80 ± 0.15	308.23 ± 2.00	0.730 ± 0.012	−30.92 ± 0.50
**F2**	100	30	5	66.27 ± 0.07	355.18 ± 4.50	0.727 ± 0.025	−30.80 ± 0.50
**F3**	100	20	10	88.54 ± 5.02	304.44 ± 3.50	0.529 ± 0.025	−34.53 ± 1.50
**F4**	100	10	15	84.82 ± 0.11	228.14 ± 3.00	0.825 ± 0.02	−33.70 ± 0.30
**F5**	100	30	15	89.34 ± 0.45	302.96 ± 2.00	0.431 ± 0.025	−36.75 ± 1.85
**F6**	125	20	5	86.73 ± 1.41	535.50 ± 2.50	0.874 ± 0.005	−36.2 ± 1.00
**F7**	125	10	10	92.28 ± 0.06	404.02 ± 2.21	0.627 ± 0.025	−34.26 ± 1.00
**F8**	125	20	10	87.79 ± 0.14	417.25 ± 4.49	0.779 ± 0.04	−34.72 ± 1.24
**F9**	125	30	10	79.77 ± 0.04	428.59 ± 5.00	0.545 ± 0.03	−37.43 ± 1.00
**F10**	125	20	15	93.98 ± 4.80	285.00 ± 5.00	0.551 ± 0.045	−34.10 ± 0.40
**F11**	150	10	5	96.34 ± 0.58	510.57 ± 1.00	0.746 ± 0.04	−41.28 ± 0.50
**F12**	150	30	5	74.53 ± 0.18	565.94 ± 5.00	0.532 ± 0.089	−40.20 ± 1.00
**F13**	150	20	10	92.54 ± 0.078	513.91 ± 5.00	0.615 ± 0.010	−39.60 ± 0.50
**F14**	**150**	**10**	**15**	**99.05 ± 0.05**	**447.15 ± 5.00**	**0.742 ± 0.008**	**−34.70 ± 0.30**
**F15**	150	30	15	93.84 ± 0.04	535.83 ± 3.50	0.751 ± 0.045	−38.25 ± 0.55

**Table 4 pharmaceutics-17-01616-t004:** Permeation results and related parameters.

Formula	Amount Permeated (µg/cm^2^)	Flux, J_ss_ (µg/cm^2^/24 h)	Permeation Coefficient, KP (cm/24 h)
**AR**	80.94 ± 4.80	11.82	0.002
**The optimum AR-loaded PEG-TERs**	37.16 ± 3.91	5.04	0.001

**Abbreviations: AR**, Aripiprazole; and **TERs**, terpesomes.

## Data Availability

Data presented in this study are contained within the article. Further inquiries can be directed to the corresponding authors.

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
