# Peer review of "PEGylated Terpesome-Loaded 3D-Printed Aripiprazole Ocuserts for the Treatment of Ocular Candidiasis"

_pharmaceutics, 2025, doi:10.3390/pharmaceutics17121616_

Round 1

Reviewer 1 Report

Comments and Suggestions for Authors

 How were the 3D-printed ocuserts characterized for drug release profiles?
What criteria were used to select aripiprazole for ocular candidiasis treatment?
How was the effectiveness of ocuserts evaluated in vitro and in vivo?
What standards were used to assess the biocompatibility of materials used?
How did the formulation influence the stability of the ocuserts?
What were the main findings regarding the treatment efficacy for ocular candidiasis?

Reviewer 2 Report

Comments and Suggestions for Authors

This study develops PEGylated terpesomes loaded with aripiprazole and incorporates them into 3D-printed PLA ocuserts for treating ocular candidiasis. The nanoformulation is optimized through a central composite design and characterized for drug loading, particle size, stability, and permeation. Ex vivo and in vivo analyses demonstrate enhanced corneal deposition and significantly superior antifungal activity compared with AR suspension. Overall, this hybrid nanocarrier, 3D-printed system represents a promising strategy for sustained ocular delivery against fungal keratitis.

The following recommendations are proposed to further improve and strengthen the manuscript:

  1. How does PEGylation specifically alter the interaction between terpesomes and corneal mucins, and did the study evaluate PEG density on surface shielding?
  2. Multiple grammatical and structural issues were noted, including missing commas after introductory phrases, overly long sentences, and inconsistent capitalization of terms such as “hydrophilic–lipophilic balance,” all of which should be revised.
  3. The manuscript contains several typographical and formatting errors, such as “amoun0074”, amount, inconsistent “ug/µg” units, and misformatted “Brji®”, that should be corrected.
  4. Certain sentences should be rephrased for precision and academic style, for example, “The optimum formulation exhibited…” could be improved to “The optimized formulation exhibited…,” and “This high stability can be due to…” should be revised to “This high stability may be attributed to…”.
  5. In the introduction add and discuss recent studies to strengthen the conversation on ocular disease mechanisms and drug-induced ocular inflammation. These references appropriately enhance the scientific context regarding natural bioactive molecules and ocular safety considerations.
  6. Add future perspectives/benefits of research in end of discussion section Thanks

Reviewer 3 Report

Comments and Suggestions for Authors

The manuscript proposes the use of aripiprazole formulated as terpesomes loaded inside a 3D-printed ocular insert for the treatment of ocular candidiasis. 

Why the terpesomes formulation was loaded into an ocular insert? Which are the advantages of the proposed formulations in comparison to the others?

The introduction should better emphasize the use of aripiprazole to treat ocular fungal infections and to discuss the existing formulations of aripiprazole for this application.

Which is the source for corneal tissue? Which is their size? How was it mounted on the Franz cell apparatus?

The PDI for the formulations are quite high (between 0.43 and 0.87). It is probably related to the preparation method using a bath sonicator rather than a probe sonicator. Does the size affect the ocular efficacy?

The mechanical properties of the 3D printed insert was not evaluated. Is it suitable for ocular administration? According to which background and how was designed the ocular insert? 

Why the ex vivo or in vivo corneal permeation of terpesome formulation loaded with aripiprazole was not investigated? This would support the relevance of the insert formulation.

Discussion with the existing literature can be improved. 

Round 2

Reviewer 1 Report

Comments and Suggestions for Authors

The article is well revised.

Reviewer 2 Report

Comments and Suggestions for Authors

The Revised Manuscript has been reviewed and authors have addressed all previous comments and suggestions and Manuscript is Acceptable in Present Form for Publication Thanks 

Reviewer 3 Report

Comments and Suggestions for Authors

The authors have addressed the reviewer comments. The manuscirpt is suitable for publication.